# Stabilizing DARTS with Amended Gradient Estimation on Architectural Parameters

## Abstract

Differentiable neural architecture search has been a popular methodology of exploring architectures for deep learning. Despite the great advantage of search efficiency, it often suffers weak stability, which hinders it from being applied to a large search space or being flexibly adjusted to different scenarios. This paper investigates DARTS, the currently most popular differentiable search algorithm, and points out an important factor of instability, which lies in its approximation on the gradients of architectural parameters. *In the current status, the optimization algorithm can converge to another point which results in dramatic inaccuracy in the re-training process.* Based on this analysis, we propose an amending term for computing architectural gradients by making use of a direct property of the optimality of network parameter optimization. Our approach mathematically guarantees that gradient estimation follows a roughly correct direction, which leads the search stage to converge on reasonable architectures. In practice, our algorithm is easily implemented and added to DARTS-based approaches efficiently. Experiments on CIFAR and ImageNet demonstrate that our approach enjoys accuracy gain and, more importantly, enables DARTS-based approaches to explore much larger search spaces that have not been studied before.

## 1 Introduction

Neural architecture search (NAS) has been an important topic in the research area of automated machine learning (AutoML). The idea is to replace the manual way of designing neural network architectures with an automatic algorithm, by which deep learning methods become more flexible in fitting complex data distributions, *e.g.*, large-scale image datasets. Early efforts of NAS involved using heuristic search methods such as reinforcement learning (Zoph & Le, 2017; Zoph et al., 2018) and evolutionary algorithms (Real et al., 2017; Xie & Yuille, 2017) to sample networks from a large search space, and optimizing each sampled network individually to evaluate its quality. Despite notable successes obtained by this methodology, it often requires a vast amount of computation, which obstacles its applications in the scenarios of limited resources. Inspired by the idea of reusing and sharing parameters among trained networks, DARTS (Liu et al., 2019b) was designed as a 'one-shot' solution of NAS. The major difference from the aforementioned methods is a differentiable formulation of architecture search and an end-to-end mechanism which optimizes model weights (such as convolution) and architectural weights simultaneously. Recently, improvements upon DARTS were made in various aspects (Chen et al., 2019; Xu et al., 2019; Liang et al., 2019), making it a reasonable tradeoff between search cost and performance.

Despite its broad applications, the current pipeline of DARTS (or, generally speaking, differentiable NAS approaches) suffers a critical weakness known as **instability**. Researchers reported (Liang et al., 2019) that DARTS-based algorithms can sometimes generate weird architectures that produce considerably worse accuracy than those generated in other individual runs, or even significantly worse than randomly generated architectures. There indeed exist tricks designed by human expertise (Chen et al., 2019; Nayman et al., 2019; Liang et al., 2019) to alleviate this issue, but we point out that **these approaches violated the ideology of NAS, which is to maximally prevent human interventions.** Moreover, even with such add-ons, a dramatic property of DARTS persists and has not been studied carefully in prior work. When DARTS, as well as its variants, gets trained for a longer time, *e.g.*, from the default number of 50 epochs to 200 epochs, we surprisingly observe that **all** these approaches converge to very similar architectures, in which almost all edges are occupied

by *skip-connect* (*a.k.a.*, *identity*). These architectures, with fewer trainable parameters, are often far from producing high accuracy in particular on large datasets like ImageNet, but they somehow produce sufficiently high validation accuracy in the search stage. **In other words, convergence in search often leads to bad performance in re-training.** This is why some previous DARTS-based approaches advocated for early termination (Liang et al., 2019), a practical but non-essential solution. Also, we conjecture that early termination also contributes to the lack of stability and, more importantly, trustfulness, of DARTS-based approaches.

This paper delves deep into the inconsistency between convergence and performance. We show that the devil lies in optimizing the loss function of the super-network, $\mathcal{L}_{\mathrm{val}}(\boldsymbol{\omega}^\star(\boldsymbol{\alpha}), \boldsymbol{\alpha})$ ($\boldsymbol{\omega}$ and $\boldsymbol{\alpha}$ denote network and architectural parameters, respectively, and $\boldsymbol{\omega}^\star(\boldsymbol{\alpha})$ is the global optimum of $\boldsymbol{\omega}$ given $\boldsymbol{\alpha}$), in which $\boldsymbol{\omega}$ and $\boldsymbol{\alpha}$ get updated alternately. Following the chain rule, $\nabla_{\boldsymbol{\alpha}}\mathcal{L}_{\mathrm{val}}(\boldsymbol{\omega}^\star(\boldsymbol{\alpha}), \boldsymbol{\alpha})|_{\boldsymbol{\alpha}=\boldsymbol{\alpha}_t}$ equals to $\nabla_{\boldsymbol{\alpha}}\mathcal{L}_{\mathrm{val}}(\boldsymbol{\omega}, \boldsymbol{\alpha})|_{\boldsymbol{\omega}=\boldsymbol{\omega}^\star(\boldsymbol{\alpha}_t), \boldsymbol{\alpha}=\boldsymbol{\alpha}_t} + \nabla_{\boldsymbol{\alpha}}\boldsymbol{\omega}^\star(\boldsymbol{\alpha})|_{\boldsymbol{\alpha}=\boldsymbol{\alpha}_t} \cdot \nabla_{\boldsymbol{\omega}}\mathcal{L}_{\mathrm{val}}(\boldsymbol{\omega}, \boldsymbol{\alpha})|_{\boldsymbol{\omega}=\boldsymbol{\omega}^\star(\boldsymbol{\alpha}_t), \boldsymbol{\alpha}=\boldsymbol{\alpha}_t}$, in which the first term is easy to compute while the second term is not, mainly because $\boldsymbol{\omega}^\star(\boldsymbol{\alpha})$ is difficult to estimate, and so is the term of $\nabla_{\boldsymbol{\alpha}}\boldsymbol{\omega}^\star(\boldsymbol{\alpha})|_{\boldsymbol{\alpha}=\boldsymbol{\alpha}_t}$. DARTS-based approaches performed inaccurate approximation for this purpose, in which the *first-order* version of DARTS directly discarded the second term – but this term is often numerically significant, and the *second-order* version of DARTS applied an approximation to this term which is not mathematically guaranteed (see Section 3.4). Consequently, the accuracy of $\nabla_{\boldsymbol{\alpha}}\mathcal{L}_{\mathrm{val}}(\boldsymbol{\omega}^\star(\boldsymbol{\alpha}), \boldsymbol{\alpha})|_{\boldsymbol{\alpha}=\boldsymbol{\alpha}_t}$ cannot be guaranteed, and hence the update of $\boldsymbol{\alpha}$ can be problematic. To the best of our knowledge, this issue is **not** studied by existing DARTS-based approaches.

To deal with this problem, we propose an alternative way of computing $\nabla_{\boldsymbol{\alpha}}\boldsymbol{\omega}^\star(\boldsymbol{\alpha})|_{\boldsymbol{\alpha}=\boldsymbol{\alpha}_t}$. We make use of an important property, *i.e.*, $\nabla_{\boldsymbol{\omega}}\mathcal{L}_{\mathrm{train}}(\boldsymbol{\omega}, \boldsymbol{\alpha})|_{\boldsymbol{\omega}=\boldsymbol{\omega}^\star(\boldsymbol{\alpha}^\dagger), \boldsymbol{\alpha}=\boldsymbol{\alpha}^\dagger} \equiv \mathbf{0}$ holds for any $\boldsymbol{\alpha}^\dagger$, which directly comes from the optimality of $\boldsymbol{\omega}^\star(\boldsymbol{\alpha})$. Differentiating both sides with respect to $\boldsymbol{\alpha}^\dagger$, we obtain a new equality which enables computing derives $\nabla_{\boldsymbol{\alpha}}\boldsymbol{\omega}^\star(\boldsymbol{\alpha})|_{\boldsymbol{\alpha}=\boldsymbol{\alpha}_t}$ with the inverse of the Hesse matrix, $\nabla_{\boldsymbol{\omega}}^2 \mathcal{L}_{\mathrm{train}}(\boldsymbol{\omega}, \boldsymbol{\alpha})|_{\boldsymbol{\omega}=\boldsymbol{\omega}^\star(\boldsymbol{\alpha}), \boldsymbol{\alpha}=\boldsymbol{\alpha}_t}$. This idea enables us to achieve a more accurate approximation on $\nabla_{\boldsymbol{\alpha}}\mathcal{L}_{\mathrm{val}}(\boldsymbol{\omega}^\star(\boldsymbol{\alpha}), \boldsymbol{\alpha})|_{\boldsymbol{\alpha}=\boldsymbol{\alpha}_t}$ when $\boldsymbol{\omega}^\star(\boldsymbol{\alpha})$ is not available. Mathematically, we prove that when we have $\boldsymbol{\omega}^{\mathrm{est}} \approx \boldsymbol{\omega}^\star(\boldsymbol{\alpha}_t)$, the inner angle between the second term and our approximate term is smaller than 90 degrees. Note that this property does **not** hold in existing DARTS-based algorithms. Our final solution involves using the amended second term of $\nabla_{\boldsymbol{\alpha}}\mathcal{L}_{\mathrm{val}}(\boldsymbol{\omega}^\star(\boldsymbol{\alpha}), \boldsymbol{\alpha})|_{\boldsymbol{\alpha}=\boldsymbol{\alpha}_t}$ meanwhile keeping the first term unchanged, which goes one step further in optimizing the super-network, which reflects in a higher validation accuracy in the search stage.

Our approach is very easily implemented. The overall computational overhead is comparable to the *second-order* version of DARTS. Experiments are performed on image classification, with popular datasets including CIFAR and ImageNet being used. In all experiments, we allow the search stage to come to a complete convergence and report competitive accuracy among current state-of-the-arts. The stability of our approach also enables us to close the gap between hyper-parameters of search and evaluation, as well as explore more complex search spaces, which are believed to be correct directions of NAS but existing DARTS-based approaches would mostly fail. Therefore, we believe our algorithm can expand the application scenario of differentiable NAS methods in particular DARTS-based approaches.

The remainder of this paper is organized as follows. We briefly review related work in Section 2, and illustrate our approach of amending architectural gradients in Section 3. After experiments are shown in Section 4, we conclude this work in Section 5.

## 2 RELATED WORK

With the era of big data and powerful computational resources, deep learning (LeCun et al., 2015), in particular, deep neural networks (Krizhevsky et al., 2012), have rapidly grown up to be the standard tool for learning representations in a complicated feature space. Recent years have witnessed the trend of using deeper (He et al., 2016) and denser (Huang et al., 2017) networks to boost recognition performance, while there is no justification that whether these manually designed architectures are best for each specific task, *e.g.*, image classification. To advance, researchers started considering the possibility of learning network architectures automatically from data, which led to the appearance

of neural architecture search (NAS) (Zoph & Le, 2017), which is now popular and known as a sub research field in automated machine learning (AutoML).

The common pipeline of NAS starts with a pre-defined space of network operators. Since the search space is often large (*e.g.*, containing $10^{10}$ or even more possible architectures), it is unlikely that exhaustive search is tractable, and thus heuristic search methods are widely applied for speedup. Typical examples include reinforcement learning (Zoph & Le, 2017; Zoph et al., 2018; Liu et al., 2018a) and evolutionary algorithms (Real et al., 2017; Xie & Yuille, 2017; Real et al., 2019). These approaches followed a general pipeline that samples a set of architectures from a learnable distribution, evaluates them and learns from rewards by updating the distribution. In an early age, each sampled architecture underwent an individual training process from scratch and thus the overall computational overhead is large, *e.g.*, hundreds of even thousands of GPU-days. To alleviate the burden, researchers started to share computation among training sampled architectures, with the key lying in reusing network weights trained previously (Cai et al., 2018) or starting from a well-trained super-network (Pham et al., 2018). These efforts shed light on the so-called one-shot architecture search methods, which required training the super-network only once and thus ran more efficiently, *e.g.*, two or three orders of magnitude faster than conventional approaches.

Within the scope of one-shot architecture search, an elegant solution lies in jointly formulating architecture search and approximation, so that it is possible to apply end-to-end optimization for training network and architectural parameters simultaneously. This methodology is known today as differentiable NAS, and a typical example is DARTS (Liu et al., 2019b), which constructed a super-network with all possible operators contained and decoupled, and the goal is to determine the weights of these architectural parameters, followed by pruning and re-training stages. This kind of approach allowed more flexible search space to be constructed, unlike conventional approaches with either reinforcement or evolutionary learning, which suffer from the computational burden and thus must constrain search within a relatively small search space (Tan & Le, 2019).

Despite the inspirations brought by differentiable NAS, these approaches still suffer a few critical issues that narrow down their applications in practice. One significant drawback lies in the lack of stability, which reflects in the way that results of differentiable search can be impacted by very small perturbations, *e.g.*, initialization of architectural weights, training hyper-parameters, and even randomness in the training process. Existing solutions include running search for several individual times and choosing the best one in validation (Liu et al., 2019b), or using other kinds of techniques such as decoupling modules (Cai et al., 2019; Guo et al., 2019), adjusting search space during optimization (Noy et al., 2019; Chen et al., 2019; Nayman et al., 2019), regularization (Xu et al., 2019), early termination (Liang et al., 2019), *etc.*, however, these approaches seemed to develop heuristic remedies rather than analyze it from the mathematical fundamentals, *e.g.*, how instability happens in mathematics.

In this paper, we investigate the stability issue in mathematics and show that the results produced by the current approaches are much less reliable than people used to think. Then, we fix this issue by amending optimization of the architectural parameters, so that each step of the update gets closer to the correct direction. We show great improvement on stability in a fundamental task, image classification, while we believe our approach can be applied to a wide range of tasks including object detection (Ghiasi et al., 2019), semantic segmentation (Liu et al., 2019a), hyper-parameter learning (Cubuk et al., 2019), *etc.*

## 3 STABILIZING DARTS WITH AMENDED GRADIENTS

In this section, we first show that DARTS can fail dramatically when it gets trained till convergence, and then we mathematically analyze how this problem is related to inaccurate approximation in optimization, following which we present our solution to amend this error and thus stabilize DARTS.

### 3.1 PRELIMINARIES: DIFFERENTIABLE NEURAL ARCHITECTURE SEARCH

Differentiable NAS approaches start with defining a super-network, which is constrained in a search space with a pre-defined number of layers and a limited set of neural operators. The core idea is to introduce a 'soft' way operator selection (*i.e.*, using a weighted sum over the outputs of a few operators instead of taking the output of only one), so that optimization can be done in an end-to-

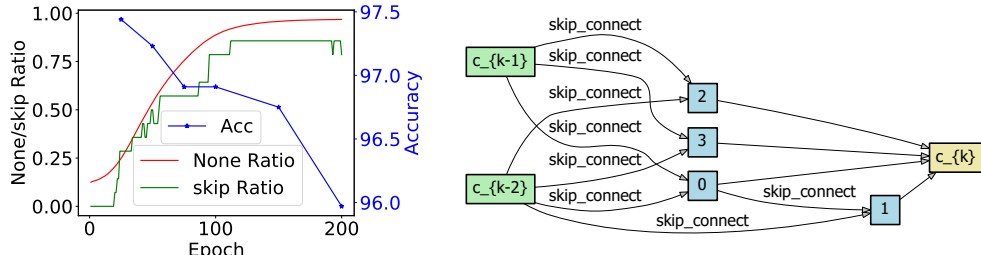

Figure 1: Left: a typical search process of the *first-order* DARTS, in which 200 search epochs are used. Red, green and blue lines indicate the average weight of *none*, the ratio of dominant *skip-connect* operators (over 14 normal edges) and the re-training accuracy, with respect to the number of search epochs, respectively. Right: the normal cell obtained after 200 search epochs, in which all preserved operators are *skip-connect*. **We executed both *first-order* and *second-order* DARTS for several times, and such failure consistently happens in each individual run.**

end manner. Mathematically, the super-network is a function $\mathbf{f}(\mathbf{x}; \boldsymbol{\omega}, \boldsymbol{\alpha})$, with $\mathbf{x}$ being input, and parameterized by network parameters $\boldsymbol{\omega}$ (*e.g.*, convolutional kernels) and architectural parameters $\boldsymbol{\alpha}$ (*e.g.*, indicating the importance of each operator between each pair of layers). $\mathbf{f}(\mathbf{x}; \boldsymbol{\omega}, \boldsymbol{\alpha})$ is differentiable to both $\boldsymbol{\omega}$ and $\boldsymbol{\alpha}$, so that gradient-based approaches can be applied for optimization.

In the example of DARTS, $\mathbf{f}(\mathbf{x}; \boldsymbol{\omega}, \boldsymbol{\alpha})$ is composed of a few cells, each of which contains $N$ nodes, and there is a pre-defined set, $\mathcal{E}$, denoting which pairs of nodes are connected. For each connected node pair $(i, j)$, $i < j$, node $j$ takes the output of node $i$, $\mathbf{x}_i$, as a part of its input, and propagate it through a pre-defined operator set, $\mathcal{O}$, with all outputs summed up:

$$\mathbf{y}^{(i,j)}(\mathbf{x}_i) = \sum_{o \in \mathcal{O}} \frac{\exp(\alpha_o^{(i,j)})}{\sum_{o' \in \mathcal{O}} \exp(\alpha_o^{(i,j)})} \cdot o(\mathbf{x}_i). \tag{1}$$

Here, a softmax term is computed by architectural weights to normalize outputs. Within each unit of the search process, $\boldsymbol{\omega}$ and $\boldsymbol{\alpha}$ get optimized alternately. After search, the operator $o$ with the maximal value of $\alpha_o^{(i,j)}$ is preserved for each edge $(i, j)$. All network parameters $\boldsymbol{\omega}$ are discarded and the obtained architecture is re-trained from scratch[1].

## 3.2 DARTS FAILS: THE CONTRADICTORY BETWEEN CONVERGENCE AND PERFORMANCE

Our research is motivated by an observation that DARTS, at the end of a regular training process with, say, 50 epochs (Liu et al., 2019b), has not yet arrived at or even got close to convergence, yet the weight of the *none* (*a.k.a.*, *zero*) operator is consistently the largest on each edge of the normal cell. To verify this, we increase the length of each training stage by 4 times, *i.e.*, from 50 to 200 epochs. Two weird phenomena are observed, both of which are shown in the left part of Figure 1. First, the weight of the *none* operator monotonically goes up – at 200 epochs, the weight has achieved 0.97 on *each* edge of the normal cells, however, this operator is not considered in the final architecture. Second, almost all preserved operators are *skip-connect* (*a.k.a.*, *identity*), a parameter-free operator that contributes little to feature learning – and surprisingly, it occupies 30% to 70% of the weight remained by *none*. Such a network has much fewer parameters than a well-designed one, and thus it usually reports unsatisfying performance at the re-training stage. This indicates that, in the context of DARTS, there exists a contradictory between search convergence and re-training accuracy.

We point out that this is a critical issue, which suggests that when using DARTS-based approaches, one does not hope the search process to achieve convergence as it implies bad performance. In other words, each 'successful' architecture comes from an early-terminated search process. Consequently, the initialization of parameters ($\boldsymbol{\alpha}$ and $\boldsymbol{\omega}$), the hyper-parameters of search (*e.g.*, learning rate) and the time of terminating search become important and thus need to be determined by experience (Liang et al., 2019). This weakens the stability as well as explainability of search and, more importantly,

---

[1]For simplicity, we ignore a lot of technical details, which we simply follow the original implementation.

violates the fundamental ideology of AutoML, *i.e.*, maximally reducing human interference and determining the best architecture by training data.

### 3.3 Delving Deep into Mathematics: Problem and Solution

We point out that the reason lies in inaccurate estimation of the gradient with respect to $\boldsymbol{\alpha}$, namely, $\nabla_{\boldsymbol{\alpha}}\mathcal{L}_{\mathrm{val}}(\boldsymbol{\omega}^{\star}(\boldsymbol{\alpha}),\boldsymbol{\alpha})|_{\boldsymbol{\alpha}=\boldsymbol{\alpha}_t}$. Following the chain rule of gradients, this quantity equals to

$$\nabla_{\boldsymbol{\alpha}}\mathcal{L}_{\mathrm{val}}(\boldsymbol{\omega},\boldsymbol{\alpha})|_{\boldsymbol{\omega}=\boldsymbol{\omega}^{\star}(\boldsymbol{\alpha}_t),\boldsymbol{\alpha}=\boldsymbol{\alpha}_t} + \nabla_{\boldsymbol{\alpha}}\boldsymbol{\omega}^{\star}(\boldsymbol{\alpha})|_{\boldsymbol{\alpha}=\boldsymbol{\alpha}_t} \cdot \nabla_{\boldsymbol{\omega}}\mathcal{L}_{\mathrm{val}}(\boldsymbol{\omega},\boldsymbol{\alpha})|_{\boldsymbol{\omega}=\boldsymbol{\omega}^{\star}(\boldsymbol{\alpha}_t),\boldsymbol{\alpha}=\boldsymbol{\alpha}_t}, \quad (2)$$

in which the first term is relatively easy to compute (as is done by the *first-order* version of DARTS), while the second term, in particular $\nabla_{\boldsymbol{\alpha}}\boldsymbol{\omega}^{\star}(\boldsymbol{\alpha})|_{\boldsymbol{\alpha}=\boldsymbol{\alpha}_t}$, is not. The *first-order* version of DARTS directly discarded this term, but it often has significant numerical values which are not negligible. The *second-order* version of DARTS indeed proposed an approximation to this term, but, as we shall see in the next subsection, can incur a large approximation error (the inner-product between the correct and estimated directions can be smaller than 0). Consequently, there can be a significant gap between the estimated and true values of $\nabla_{\boldsymbol{\alpha}}\mathcal{L}_{\mathrm{val}}(\boldsymbol{\omega}^{\star}(\boldsymbol{\alpha}),\boldsymbol{\alpha})|_{\boldsymbol{\alpha}=\boldsymbol{\alpha}_t}$. Such inaccuracy accumulates with every update on $\boldsymbol{\alpha}$, and gradually causes $\boldsymbol{\alpha}$ to converge to weird solutions that are far from optimum, *e.g.*, the entire super-network is dominated by *none* and *skip-connect* operators. We name this phenomenon as the **gradient trap** during optimization.

To estimate $\nabla_{\boldsymbol{\alpha}}\boldsymbol{\omega}^{\star}(\boldsymbol{\alpha})|_{\boldsymbol{\alpha}=\boldsymbol{\alpha}_t}$, we first make a reasonable assumption that $\nabla_{\boldsymbol{\alpha}}\boldsymbol{\omega}^{\star}(\boldsymbol{\alpha})|_{\boldsymbol{\alpha}=\boldsymbol{\alpha}_t}$ has finite values. Then, we make use of an important property, *i.e.*, $\nabla_{\boldsymbol{\omega}}\mathcal{L}_{\mathrm{train}}(\boldsymbol{\omega},\boldsymbol{\alpha})|_{\boldsymbol{\omega}=\boldsymbol{\omega}^{\star}(\boldsymbol{\alpha}^{\dagger}),\boldsymbol{\alpha}=\boldsymbol{\alpha}^{\dagger}} \equiv \mathbf{0}$ holds for any $\boldsymbol{\alpha}^{\dagger}$. This is property directly comes from the optimality of $\boldsymbol{\omega}^{\star}(\boldsymbol{\alpha})$, but it has never been used by existing approaches. Applying differentiation with respect to any $\boldsymbol{\alpha}^{\dagger}$ to both sides of this equality, we have $\nabla_{\boldsymbol{\alpha}^{\dagger}}\left(\nabla_{\boldsymbol{\omega}}\mathcal{L}_{\mathrm{train}}(\boldsymbol{\omega},\boldsymbol{\alpha})|_{\boldsymbol{\omega}=\boldsymbol{\omega}^{\star}(\boldsymbol{\alpha}^{\dagger}),\boldsymbol{\alpha}=\boldsymbol{\alpha}^{\dagger}}\right) \equiv \mathbf{0}$. When $\boldsymbol{\alpha}^{\dagger} = \boldsymbol{\alpha}_t$, it becomes:

$$\nabla_{\boldsymbol{\alpha}^{\dagger}}\left(\nabla_{\boldsymbol{\omega}}\mathcal{L}_{\mathrm{train}}(\boldsymbol{\omega},\boldsymbol{\alpha})|_{\boldsymbol{\omega}=\boldsymbol{\omega}^{\star}(\boldsymbol{\alpha}^{\dagger}),\boldsymbol{\alpha}=\boldsymbol{\alpha}^{\dagger}}\right)\Big|_{\boldsymbol{\alpha}^{\dagger}=\boldsymbol{\alpha}_t} = \mathbf{0}. \quad (3)$$

Again, applying the chain rule to the left-hand side gives:

$$\nabla_{\boldsymbol{\alpha},\boldsymbol{\omega}}^{2}\mathcal{L}_{\mathrm{train}}(\boldsymbol{\omega},\boldsymbol{\alpha})\big|_{\boldsymbol{\omega}=\boldsymbol{\omega}^{\star}(\boldsymbol{\alpha}_t),\boldsymbol{\alpha}=\boldsymbol{\alpha}_t} + \nabla_{\boldsymbol{\alpha}}\boldsymbol{\omega}^{\star}(\boldsymbol{\alpha})\big|_{\boldsymbol{\alpha}=\boldsymbol{\alpha}_t} \cdot \nabla_{\boldsymbol{\omega}}^{2}\mathcal{L}_{\mathrm{train}}(\boldsymbol{\omega},\boldsymbol{\alpha})\big|_{\boldsymbol{\omega}=\boldsymbol{\omega}^{\star}(\boldsymbol{\alpha}_t),\boldsymbol{\alpha}=\boldsymbol{\alpha}_t} = \mathbf{0}, \quad (4)$$

where we use the notation $\nabla_{\boldsymbol{\alpha},\boldsymbol{\omega}}^{2}(\cdot) \equiv \nabla_{\boldsymbol{\alpha}}(\nabla_{\boldsymbol{\omega}}(\cdot))$ throughout the remaining part of this paper. Here, $\nabla_{\boldsymbol{\omega}}^{2}\mathcal{L}_{\mathrm{train}}(\boldsymbol{\omega},\boldsymbol{\alpha})\big|_{\boldsymbol{\omega}=\boldsymbol{\omega}^{\star}(\boldsymbol{\alpha}_t),\boldsymbol{\alpha}=\boldsymbol{\alpha}_t} \doteq \mathbf{H}$ is the Hesse matrix corresponding to the optimum $\boldsymbol{\omega}^{\star}(\boldsymbol{\alpha}_t)$, which is symmetric and positive-definite, and thus invertible. This gives us an estimation that $\nabla_{\boldsymbol{\alpha}}\boldsymbol{\omega}^{\star}(\boldsymbol{\alpha})|_{\boldsymbol{\alpha}=\boldsymbol{\alpha}_t} = -\nabla_{\boldsymbol{\alpha},\boldsymbol{\omega}}^{2}\mathcal{L}_{\mathrm{train}}(\boldsymbol{\omega},\boldsymbol{\alpha})\big|_{\boldsymbol{\omega}=\boldsymbol{\omega}^{\star}(\boldsymbol{\alpha}_t),\boldsymbol{\alpha}=\boldsymbol{\alpha}_t} \cdot \mathbf{H}^{-1}$. Substituting it into Equation 2 gives:

$$\nabla_{\boldsymbol{\alpha}}\mathcal{L}_{\mathrm{val}}(\boldsymbol{\omega}^{\star}(\boldsymbol{\alpha}),\boldsymbol{\alpha})|_{\boldsymbol{\alpha}=\boldsymbol{\alpha}_t} = \nabla_{\boldsymbol{\alpha}}\mathcal{L}_{\mathrm{val}}(\boldsymbol{\omega},\boldsymbol{\alpha})|_{\boldsymbol{\omega}=\boldsymbol{\omega}^{\star}(\boldsymbol{\alpha}_t),\boldsymbol{\alpha}=\boldsymbol{\alpha}_t} -$$
$$\nabla_{\boldsymbol{\alpha},\boldsymbol{\omega}}^{2}\mathcal{L}_{\mathrm{train}}(\boldsymbol{\omega},\boldsymbol{\alpha})\big|_{\boldsymbol{\omega}=\boldsymbol{\omega}^{\star}(\boldsymbol{\alpha}_t),\boldsymbol{\alpha}=\boldsymbol{\alpha}_t} \cdot \mathbf{H}^{-1} \cdot \nabla_{\boldsymbol{\omega}}\mathcal{L}_{\mathrm{val}}(\boldsymbol{\omega},\boldsymbol{\alpha})|_{\boldsymbol{\omega}=\boldsymbol{\omega}^{\star}(\boldsymbol{\alpha}_t),\boldsymbol{\alpha}=\boldsymbol{\alpha}_t}. \quad (5)$$

Note that Equation 5 does not involve any approximation. The only issue comes from the term of $\mathbf{H}^{-1}$, which is computationally intractable due to the large dimensionality of $\mathbf{H}$ (it is related to the number of network parameters, which often exceeds one million in a typical super-network).

### 3.4 Approximations in Computing the Inverse Hesse Matrix

Let us denote Equation 5 in an abbreviated form of $\mathbf{g} = \mathbf{g}_1 + \mathbf{g}_2$, in which $\mathbf{g}_1$, the *first-order* term of DARTS, is easily computed, while $\mathbf{g}_2$ is not due to the computation of $\mathbf{H}^{-1}$. Here, we propose an alternative solution which constructs an approximation term $\mathbf{g}_2'$:

$$\mathbf{g}_2' = -\eta \cdot \nabla_{\boldsymbol{\alpha},\boldsymbol{\omega}}^{2}\mathcal{L}_{\mathrm{train}}(\boldsymbol{\omega},\boldsymbol{\alpha})\big|_{\boldsymbol{\omega}=\boldsymbol{\omega}^{\star}(\boldsymbol{\alpha}_t),\boldsymbol{\alpha}=\boldsymbol{\alpha}_t} \cdot \mathbf{H} \cdot \nabla_{\boldsymbol{\omega}}\mathcal{L}_{\mathrm{val}}(\boldsymbol{\omega},\boldsymbol{\alpha})|_{\boldsymbol{\omega}=\boldsymbol{\omega}^{\star}(\boldsymbol{\alpha}_t),\boldsymbol{\alpha}=\boldsymbol{\alpha}_t}, \quad (6)$$

where $\eta > 0$ is named the *amending coefficient*. In what follows, we show that $\mathbf{g}_2'$ is indeed a reasonable approximation of $\mathbf{g}_2$. Since

$$\langle\mathbf{g}_2',\mathbf{g}_2\rangle = \eta \cdot \nabla_{\boldsymbol{\omega}}\mathcal{L}_{\mathrm{val}}(\boldsymbol{\omega},\boldsymbol{\alpha})|_{\boldsymbol{\omega}=\boldsymbol{\omega}^{\star}(\boldsymbol{\alpha}_t),\boldsymbol{\alpha}=\boldsymbol{\alpha}_t}^{\top} \cdot \mathbf{H}^{-1} \cdot \nabla_{\boldsymbol{\omega},\boldsymbol{\alpha}}^{2}\mathcal{L}_{\mathrm{train}}(\boldsymbol{\omega},\boldsymbol{\alpha})\big|_{\boldsymbol{\omega}=\boldsymbol{\omega}^{\star}(\boldsymbol{\alpha}_t),\boldsymbol{\alpha}=\boldsymbol{\alpha}_t} \cdot$$
$$\nabla_{\boldsymbol{\alpha},\boldsymbol{\omega}}^{2}\mathcal{L}_{\mathrm{train}}(\boldsymbol{\omega},\boldsymbol{\alpha})\big|_{\boldsymbol{\omega}=\boldsymbol{\omega}^{\star}(\boldsymbol{\alpha}_t),\boldsymbol{\alpha}=\boldsymbol{\alpha}_t} \cdot \mathbf{H} \cdot \nabla_{\boldsymbol{\omega}}\mathcal{L}_{\mathrm{val}}(\boldsymbol{\omega},\boldsymbol{\alpha})|_{\boldsymbol{\omega}=\boldsymbol{\omega}^{\star}(\boldsymbol{\alpha}_t),\boldsymbol{\alpha}=\boldsymbol{\alpha}_t}, \quad (7)$$

the product of the two terms between $\mathbf{H}^{-1}$ and $\mathbf{H}$ is a semi-positive-definite matrix, and so is the matrix after similarity transformation, which directly gives $\langle \mathbf{g}_2', \mathbf{g}_2 \rangle \geqslant 0$.

In summary, we decompose the gradient of architectural parameters into two terms, $\mathbf{g}_1$ and $\mathbf{g}_2$, compute $\mathbf{g}_1$ directly and use an approximation to $\mathbf{g}_2$ so that the angle between the accurate and approximated terms is smaller than 90 degrees. In comparison, existing DARTS-based approaches either discarded $\mathbf{g}_2$ entirely or used a mathematically non-explainable approximation $\mathbf{g}_2'' = -\eta \cdot \nabla_{\boldsymbol{\alpha},\boldsymbol{\omega}}^2 \mathcal{L}_{\text{train}}(\boldsymbol{\omega},\boldsymbol{\alpha})\big|_{\boldsymbol{\omega}=\boldsymbol{\omega}^\star(\boldsymbol{\alpha}_t),\boldsymbol{\alpha}=\boldsymbol{\alpha}_t} \cdot \mathbf{I} \cdot \nabla_{\boldsymbol{\omega}} \mathcal{L}_{\text{val}}(\boldsymbol{\omega},\boldsymbol{\alpha})\big|_{\boldsymbol{\omega}=\boldsymbol{\omega}^\star(\boldsymbol{\alpha}_t),\boldsymbol{\alpha}=\boldsymbol{\alpha}_t}$. None of them are reasonable because $\mathbf{g}_2$ can be large, yet there is no guarantee that $\langle \mathbf{g}_2'', \mathbf{g}_2 \rangle \geqslant 0$, *i.e.*, the *second-order* DARTS can lead the algorithm to a wrong direction.

The rationality of our approach is also verified by the validation process, *i.e.*, in updating architectural parameters. The *first-order* DARTS, by directly discarding $\mathbf{g}_2$, reported an average validation accuracy of $90.5\%$ in search space $\mathcal{S}_1$ (see Section 4.1.3) on the CIFAR10 dataset. The *second-order* DARTS added $\mathbf{g}_2''$, which has no guarantee that $\langle \mathbf{g}_2'', \mathbf{g}_2 \rangle \geqslant 0$, and thus resulted in a reduced validation accuracy. Our approach, by adding $\mathbf{g}_2'$, achieves a validation accuracy of $91.5\%$, implying that our optimization works better than DARTS. This eventually results in the advantage of searched architectures, which will be verified in Section 4.1.2.

The remainder part of computing $\mathbf{g}_2'$ simply follows conventions, which we replace $\boldsymbol{\omega}^\star(\boldsymbol{\alpha})$ with the current $\boldsymbol{\omega}^{\text{est}}$ as the most accurate approximation we can get[2]. Computing $\mathbf{g}_2'$ with Equation 6 requires both $\nabla_{\boldsymbol{\alpha},\boldsymbol{\omega}}^2 \mathcal{L}_{\text{train}}(\boldsymbol{\omega},\boldsymbol{\alpha})\big|_{\boldsymbol{\omega}=\boldsymbol{\omega}^\star(\boldsymbol{\alpha}_t),\boldsymbol{\alpha}=\boldsymbol{\alpha}_t}$ and $\nabla_{\boldsymbol{\omega}} \mathcal{L}_{\text{val}}(\boldsymbol{\omega},\boldsymbol{\alpha})\big|_{\boldsymbol{\omega}=\boldsymbol{\omega}^\star(\boldsymbol{\alpha}_t),\boldsymbol{\alpha}=\boldsymbol{\alpha}_t}$, while DARTS needs the former one with $\boldsymbol{\omega}^\star(\boldsymbol{\alpha}_t)$ estimated in two steps. Therefore, computing Equation 6 requires similar computational overhead compared to the *second-order* version of DARTS. In experiments, each search epoch requires around 0.02 GPU-days on the standard 8-cell space on CIFAR10.

## 3.5 DISCUSSIONS AND RELATIONSHIP TO PRIOR WORK

The core benefit brought by our approach is the consistency between search and evaluation. This is indeed a fundamental idea of NAS, but it was ignored by existing approaches since they have been perplexed by a more significant error caused by inaccurate optimization. After the error, we point out a few prior conventions that need to be adjusted accordingly, including using different depths (*e.g.*, DARTS used 8 cells in search and 20 cells in evaluation) and widths (*e.g.*, DARTS used a basic channel number of 16 in search and 36 in evaluation) during search and evaluation, as well as using different training strategies (*e.g.*, during re-training, a few regularization techniques including Cutout (DeVries & Taylor, 2017), Dropout (Srivastava et al., 2014) and auxiliary loss were used, but none of them appeared in search). More importantly, the search process was followed by edge removal (8 out of 14 connections were preserved) which caused a significant difference between the network architectures of search and evaluation. Our approach provides the opportunity to bridge the gap between search and evaluation, which we will show in Section 4.1.2 that unifying these hyper-parameters leads to better performance.

A few prior differentiable search approaches noticed the issue of instability, but they chose to solve it in different manners. For example, P-DARTS (Chen et al., 2019) fixed the number of preserved *skip-connect* operators, PC-DARTS (Xu et al., 2019) used edge normalization to eliminate the *none* operator, while XNAS (Nayman et al., 2019) and DARTS+ (Liang et al., 2019) introduced a few human expertise to stabilize search. However, we point out that (i) either P-DARTS or PC-DARTS, with carefully designed methods or tricks, can also fail in a sufficiently long search process (more than 200 epochs); and that (ii) XNAS and DARTS+, by adding human expertise, somewhat violated the design nature of AutoML, in which one is expected to avoid introducing too many hand-designed rules.

Another line of NAS, besides differentiable methods, is to use either reinforcement learning or an evolutionary algorithm as a controller of heuristic search and train each sampled network to get some kind of rewards, *e.g.*, validation accuracy. In the viewpoint of optimization, this pipeline mainly differs from the differentiable one in that optimizing $\boldsymbol{\alpha}$ is decoupled from optimizing $\boldsymbol{\omega}$, so

---

[2]This raises another benefit of using $\mathbf{g}_2'$ to approximate $\mathbf{g}_2$. Note that $\mathbf{g}_2$ (Equation 5) requires computing $\mathbf{H}^{-1}$ while $\mathbf{g}_2'$ (Equation 6) only requires $\mathbf{H}$. The existence of $\mathbf{H}^{-1}$ is only guaranteed at $\boldsymbol{\omega} = \boldsymbol{\omega}^\star(\boldsymbol{\alpha})$, and thus the estimation can be quite inaccurate even if $\boldsymbol{\omega}^{\text{est}}$ gets sufficiently close to $\boldsymbol{\omega}^\star(\boldsymbol{\alpha})$. Therefore, using $\mathbf{H}$ itself is more friendly in this scenario.

that it does not require $\boldsymbol{\omega}$ to arrive at $\boldsymbol{\omega}^*$, but only need a reasonable approximation of $\boldsymbol{\omega}^*$ to predict model performance – this is an important reason that such algorithms often produce stable results. Our approach sheds light on introducing a similar property, *i.e.*, robustness to approximated $\boldsymbol{\omega}^*$, which helps in stabilizing differentiable search approaches.

## 4 EXPERIMENTS

### 4.1 RESULTS ON CIFAR10

The CIFAR10 dataset (Krizhevsky & Hinton, 2009) has 50,000 training and 10,000 testing images, sized $32 \times 32$, and equally distributed over 10 classes. We mainly use this dataset to evaluate the stability of our approach, as well as analyze the impacts of different search options and parameters.

#### 4.1.1 IMPACT OF THE AMENDING COEFFICIENT

We first investigate how the amending coefficient, $\eta$, defined in Equation 6, impacts architecture search. We search and re-train similarly as DARTS. During the search, all operators are assigned equal weights on each edge. We use a base channel number of 16, and a batch size of 96. An Adam optimizer is used to update architectural parameters, with a learning rate of 0.0003, a weight decay of 0.001 and a momentum of $(0.5, 0.999)$. The number of epochs is to be discussed later. During re-training, the base channel number is increased to 36. An SGD optimizer is used with an initial learning rate of 0.025, decaying following the cosine annealing rule and arriving at 0 after 600 epochs. The weight decay is set to be 0.0003, and the momentum is 0.9.

To arrive at convergence, we run the search stage for 500 epochs. We evaluate different $\eta$ values from 0 to 1, and the architectures corresponding to $\eta = 0$ (equivalent to DARTS), $\eta = 0.1$ and $\eta = 1$ are summarized in Figure 2. We can see that $\eta = 0.1$ converges, after 500, into a reasonable architecture that achieves an error rate of 3.08% on CIFAR10. We emphasize that, even with more search epochs, this architecture is not likely to change, as the preserved operator on each edge has a weight not smaller than 0.5, and most of these weights are still growing gradually.

When $\eta$ is very small, *e.g.*, $\eta = 0.001$ or $\eta = 0.01$, the change brought by this amending term to architecture search is ignorable, and our approach shows almost the same behavior as the *first-order* version of DARTS, *i.e.*, $\eta = 0$. In addition, when $\eta$ grows up, *e.g.*, from 0.001 to 0.01, although the search process eventually runs into an architecture with all *skip-connect* operators, the number of epochs needed for a complete failure is significantly postponed, which verifies that the amending term indeed pulls architecture search away from the gradient trap.

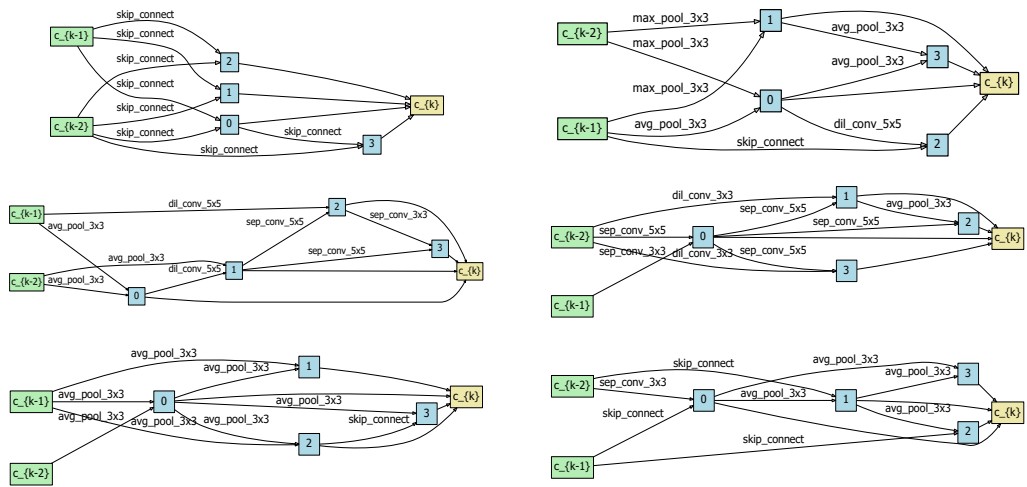

Figure 2: Normal (left) and reduction (right) cells (the standard DARTS space) obtained by different amending coefficients, namely, $\eta = 0$ (top), $\eta = 0.1$ (middle) and $\eta = 1$ (bottom).

On the other hand, if we use a sufficiently large $\eta$ value, *e.g.*, $\eta = 1$, the amending term, $\mathbf{g}'_2$, can dominate optimization, so that the first term, *i.e.*, the gradient of architectural parameters, has limited effects in updating $\boldsymbol{\alpha}$. Note that the amending term is closely related to network regularization, therefore, in the scenarios of a large $\eta$, the network significantly prefers *avg-pooling* to other operators, as *avg-pooling* can smooth feature maps and avoid over-fitting. However, *avg-pooling* is also a parameter-free operator, so the performance of such architectures is also below satisfaction.

Following these analyses, we simply use $\eta = 0.1$ for all later experiments. We do not tune $\eta$ very carefully, though it is possible to determine $\eta$ automatically using a held-out validation set. Besides, we find that the best architecture barely changes after 100 search epochs, which we fix the total length to be 100 epochs in all remaining experiments.

### 4.1.2 CONSISTENCY BETWEEN SEARCH AND EVALUATION

As discussed in Section 3.5, it is important to alleviate the difference between search and evaluation. We make the following modifications, listed from most to least important. First, to avoid edge removal, we fix the edges in each cell, so that each node $i$ is connected to node $i - 1$ and the least indexed node (denoted by $c_{k-2}$ in most conventions), resulting in 8 edges in each cell. Note that our approach also works well with all 14 edges preserved, but we have used 8 edges to be computationally fair to DARTS. Second, we unify the width (the number of basic channels) as 36 for both search and evaluation. Third, we add normalization techniques, including Cutout (DeVries & Taylor, 2017), Dropout (Srivastava et al., 2014) and an auxiliary loss tower, into the search stage.

We use the amending coefficient $\eta = 0.1$ learned from previous experiments, *i.e.*, without modification, the searched architecture, denoted by $\mathbb{A}_{\mathrm{orig}}$, is shown in the middle column of Figure 2. After modification, the obtained architecture, denoted by $\mathbb{A}_{\mathrm{new}}$, is shown in Figure 3. We re-train both networks on CIFAR10, with or without the option that stacking duplicate cells to make the network deeper (with 20 cells). With a standard re-training process, $\mathbb{A}_{\mathrm{orig}}$ reports a $3.67\%$ error with 8 cells, and a $3.08\%$ error with 20 cells; and the corresponding numbers are $3.20\%$ and $2.81\%$ for $\mathbb{A}_{\mathrm{new}}$. We find that $\mathbb{A}_{\mathrm{new}}$ is consistently better than $\mathbb{A}_{\mathrm{orig}}$, which suggests that alleviating the gap indeed helps. This also reminds us of the significant *depth gap* (Chen et al., 2019) between search and re-training (the network has 8 cells in search, but 20 cells in re-training), and this gap also obstructs our approach from achieving better performance. We will investigate this issue in the following part.

### 4.1.3 EXPLORING MORE COMPLEX SEARCH SPACES

We note tremendous efforts made by existing approaches to alleviate the *depth gap*, while our solution is a direct one, thanks to the stability of our approach which enables us to directly explore larger search spaces. Here, we denote the original search space used in DARTS as $\mathcal{S}_1$, which has six normal cells and two reduction cells, and all normal cells share the same set of architectural parameters and so do the reduction cells. Note that we have fixed the edges in this space, resulting in the total number of possible architectures to reduce from $1.1 \times 10^{18}$ to $3.3 \times 10^{13}$. We also explore a more complex search space, denoted by $\mathcal{S}_2$, in which we relax the constraint that either normal cells or reduction cells should be the same, and also the number of cells increases from 8 to 20, to be applied in re-training. Here, limited by GPU memory, we cannot support all seven operators to be searched, so we only choose two, namely *skip-connect* and *sep-conv-3x3*, which have very different properties. This setting allows a total of $1.5 \times 10^{48}$ architectures to appear, much larger than $\mathcal{S}_1$.

Results are listed in Table 1. In $\mathcal{S}_1$, our approach achieves a moderate error rate of $2.81\%$, mainly because the assumption of consistency between search and re-training does not hold. In $\mathcal{S}_2$, with directly searching in deep architectures, our result is significantly boosted (an error rate of $2.60\%$, the architecture is shown in the middle row of Figure 3). Again, we emphasize that we report re-training results based on an converged architecture, which stands out from existing DARTS-based approaches which required early termination.

In $\mathcal{S}_2$, we compare our approach against both DARTS (no amending term) and random search. We observe that DARTS produces weird architectures (the bottom row of Figure 3), that high-layer cells are mostly occupied by *skip-connect*, which is not likely to fully utilize the ability of the super-network. Regarding random search, we follow DARTS by randomly sampling 20 valid architectures from each of $\mathcal{S}_1$ and $\mathcal{S}_2$, and using a small validation dataset to choose the best two architectures for re-training. Given a fixed number of probes, it becomes more and more difficult to sufficiently

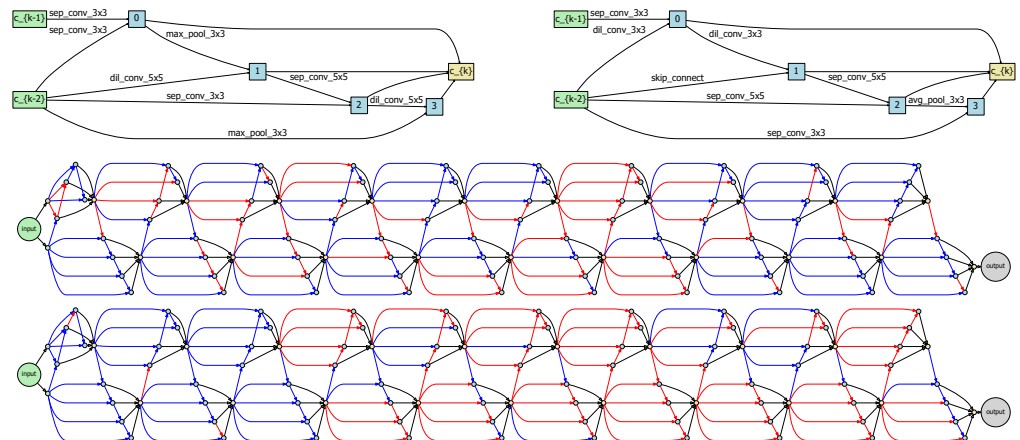

Figure 3: Top: normal and reduction cells found in $\mathcal{S}_1$. Middle & Bottom: the overall architecture found in $\mathcal{S}_2$ by DARTS, with and without the amended term, in which red and blue edges indicate *skip-connect* and *sep-conv-3x3* operators, respectively.

explore a large space. The deficits of DARTS and random search on CIFAR10 are $0.25\%$ and $0.29\%$, respectively, which do not seem to be very large arguably because CIFAR10 is relatively easy and our edge-fixed search space guarantees sufficient depth. However, when we transfer these architectures to ImageNet, the deficits become much larger, *i.e.*, with $1.7\%$ and $0.8\%$ top-1 accuracy drops, respectively (DARTS produces inferior performance to random search). These results remind us of prior work (Xie et al., 2019) which claimed that random search works sufficiently well in large search spaces. Here, we leave a comment on this debate, demonstrating that a large space indeed raises challenges to architecture search, but a stabilized search algorithm still has the ability of to find more powerful architectures.

### 4.1.4 COMPARISON TO THE STATE-OF-THE-ARTS

Finally, we compare our approach with recent approaches, in particular, differentiable ones. Result are shown in Table 1. Our approach produces competitive results among state-of-the-arts, although it does not seem to beat others. We note that existing approaches often used additional tricks, *e.g.*, P-DARTS assumed a fixed number of *skip-connect* operators, which shrinks the search space (so as to guarantee stability). More importantly, all these differentiable search approaches must be terminated in an early stage, which makes them less convincing as search has not arrived at convergence. These

| Architecture | Test Err. | Params | Search Cost | Search Method |
|---|---|---|---|---|
| | (%) | (M) | (GPU-days) | |
| DenseNet-BC (Huang et al., 2017) | 3.46 | 25.6 | - | manual |
| ENAS (Pham et al., 2018) w/ Cutout | 2.89 | 4.6 | 0.5 | RL |
| NASNet-A (Zoph et al., 2018) w/ Cutout | 2.65 | 3.3 | 1800 | RL |
| NAONet-WS (Luo et al., 2018) | 3.53 | 3.1 | 0.4 | NAO |
| Hireachical Evolution (Liu et al., 2018b) | 3.75±0.12 | 15.7 | 300 | evolution |
| AmoebaNet-B (Real et al., 2019) w/ Cutout | 2.55±0.05 | 2.8 | 3150 | evolution |
| PNAS (Liu et al., 2018a) | 3.41±0.09 | 3.2 | 225 | SMBO |
| DARTS (first-order) (Liu et al., 2019b) w/ Cutout | 3.00±0.14 | 3.3 | 0.4 | gradient-based |
| DARTS (second-order) (Liu et al., 2019b) w/ Cutout | 2.76±0.09 | 3.3 | 1.0 | gradient-based |
| SNAS (moderate) (Xie et al., 2018) w/ Cutout | 2.85±0.02 | 2.8 | 1.5 | gradient-based |
| ProxylessNAS (Cai et al., 2019) w/ Cutout | 2.08 | - | 4.0 | gradient-based |
| P-DARTS (Chen et al., 2019) w/ Cutout | 2.50 | 3.4 | 0.3 | gradient-based |
| BayesNAS (Zhou et al., 2019) w/ Cutout | 2.81±0.04 | 3.4 | 0.2 | gradient-based |
| PC-DARTS (Xu et al., 2019) w/ Cutout | 2.57±0.07 | 3.6 | 0.1 | gradient-based |
| Amended-DARTS, $\mathcal{S}_1$, w/ Cutout | 2.81±0.21 | 3.5 | 1.0 | gradient-based |
| Amended-DARTS, $\mathcal{S}_2$, w/ Cutout | 2.60±0.15 | 3.6 | 1.1 | gradient-based |

Table 1: Comparison with state-of-the-art network architectures on CIFAR10.

| Architecture | Test Err. (%) | | Params | $\times+$ | Search Cost | Search Method |
|---|---|---|---|---|---|---|
| | top-1 | top-5 | (M) | (M) | (GPU-days) | |
| Inception-v1 (Szegedy et al., 2015) | 30.2 | 10.1 | 6.6 | 1448 | - | manual |
| MobileNet (Howard et al., 2017) | 29.4 | 10.5 | 4.2 | 569 | - | manual |
| ShuffleNet $2\times$ (v1) (Zhang et al., 2018) | 26.4 | 10.2 | $\sim$5 | 524 | - | manual |
| ShuffleNet $2\times$ (v2) (Ma et al., 2018) | 25.1 | - | $\sim$5 | 591 | - | manual |
| NASNet-A (Zoph et al., 2018) | 26.0 | 8.4 | 5.3 | 564 | 1800 | RL |
| MnasNet-92 (Tan et al., 2019) | 25.2 | 8.0 | 4.4 | 388 | - | RL |
| PNAS (Liu et al., 2018a) | 25.8 | 8.1 | 5.1 | 588 | 225 | SMBO |
| AmoebaNet-C (Real et al., 2019) | 24.3 | 7.6 | 6.4 | 570 | 3150 | evolution |
| DARTS (second-order) (Liu et al., 2019b) | 26.7 | 8.7 | 4.7 | 574 | 4.0 | gradient-based |
| SNAS (mild) (Xie et al., 2018) | 27.3 | 9.2 | 4.3 | 522 | 1.5 | gradient-based |
| BayesNAS (Zhou et al., 2019) | 26.5 | 8.9 | 3.9 | - | 0.2 | gradient-based |
| P-DARTS (CIFAR10) (Chen et al., 2019) | 24.4 | 7.4 | 4.9 | 557 | 0.3 | gradient-based |
| ProxylessNAS (GPU)[‡] (Cai et al., 2019) | 24.9 | 7.5 | 7.1 | 465 | 8.3 | gradient-based |
| PC-DARTS (Xu et al., 2019)[‡] | 24.2 | 7.3 | 5.3 | 597 | 3.8 | gradient-based |
| DARTS+ (Liang et al., 2019)[‡] | 23.9 | 7.4 | 5.1 | 582 | 6.8 | gradient-based |
| Amended-DARTS, $\mathcal{S}_2$ | 24.3 | 7.4 | 5.5 | 590 | 1.1 | gradient-based |

Table 2: Comparison with state-of-the-art architectures on ILSVRC2012. All searched architectures are fit into the *mobile setting*. [‡] indicates architectures searched on ImageNet.

tricks somewhat violate the ideology of neural architecture search; in comparison, our research, though not producing the best performance, seems going along a correct and promising direction.

## 4.2 RESULTS ON ILSVRC2012

ILSVRC2012 (Russakovsky et al., 2015) is the most commonly used subset of ImageNet (Deng et al., 2009). It contains $1.3$M training images and $50$K testing images, which are almost evenly distributed over all $1,000$ categories. We directly use the $\mathcal{S}_2$ architecture obtained from CIFAR10 experiments and enlarge it with a basic number of channels of $42$, so that the FLOPs of our model is $590$M, *i.e.*, fitting the mobile setting. During re-training, there are a total of $250$ epochs. We use an SGD optimizer with an initial learning rate of $0.5$ (decaying linearly after each epoch), a momentum of $0.9$ and a weight decay of $3 \times 10^{-5}$. On NVIDIA Tesla V100 GPUs, the entire re-training process takes around 24 GPU-days.

The comparison of our approach to existing approaches is shown in Table 2. Our approach achieves a top-1 error rate of $24.3\%$ without any common optimization tricks such as AutoAugment (Cubuk et al., 2019) and Squeeze-and-Excitation modules (Hu et al., 2018). This result is competitive among state-of-the-arts, and it is obtained after convergence is achieved in the search stage. On the other hand, without the amending term, DARTS converges to a solution that *skip-connected* and *sep-conv-3x3* are largely separated, on which the re-training performance is even inferior to random search.

## 5 CONCLUSIONS

In this paper, we present an effective approach for stabilizing differentiable neural architecture search. Our motivation comes from that DARTS-based approaches mostly generate all-*skip-connect* architectures when they are executed for a sufficient number of epochs. We analyze this weird phenomenon mathematically and find the reason to lie in the dramatic inaccuracy in estimating gradients of architectural parameters. With an alternative approximation based on the optimality of network parameters, we guarantee the update of architectural parameters to be in a correct direction. In DARTS-based search spaces on CIFAR10 and ImageNet, our approach shows improved stability, in particular in large search spaces, as well as improved performance in the re-training stage.

Our research sheds light on future research on NAS in several aspects. First, we reveal that previous differentiable approaches were mostly built upon a dangerous pipeline, and mostly introduced heavy human expertise to avoid failure. By fixing the 'system error' of this pipeline, we provide a platform that NAS approaches can compete in the ability of NAS. Second, our approach enables researchers to explore even bigger search spaces that have not been studied before (due to search instability).

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

# A   APPENDIX

## A.1   A TOY EXAMPLE TO SHOW THE IMPORTANCE OF $\mathbf{g}_2$

Let the loss function be $\mathcal{L}(\omega, \alpha; x) = (\omega x - \alpha)^2$. Then, the only difference between $\mathcal{L}_{\text{train}}(\omega, \alpha) = \mathcal{L}(\omega, \alpha; x_{\text{train}})$ and $\mathcal{L}_{\text{valid}}(\omega, \alpha) = \mathcal{L}(\omega, \alpha; x_{\text{valid}})$ lies in the input, $x$. Assume the input of training data is $x_{\text{train}} = 1$ and the input validation data is $x_{\text{valid}} = 2$. It is easy to derive that the local optimum of $\mathcal{L}_{\text{train}}(\omega, \alpha)$ is $\omega^*(\alpha) = \alpha$. Substituting $x_{\text{valid}} = 2$ into $\mathcal{L}_{\text{valid}}(\omega, \alpha)$ yields $\mathcal{L}_{\text{valid}}(\omega, \alpha) = (2\omega - \alpha)^2$. When $\alpha = \alpha_{\text{t}}$, $\omega$ arrives at $\omega^*(\alpha_{\text{t}})$, so $\mathbf{g}_1 = 2(\alpha_{\text{t}} - 2\alpha_{\text{t}}) = -2\alpha_{\text{t}}$ and $\mathbf{g}_2 = 4\alpha_{\text{t}}$. When $\omega$ arrives at $\omega^*(\alpha)$, $\mathcal{L}_{valid}(\omega^*(\alpha), \alpha) = \alpha^2$, $\mathbf{g}(\alpha_{\text{t}}) = 2\alpha_{\text{t}} = \mathbf{g}_1 + \mathbf{g}_2$. In summary, both $\mathbf{g}_1$ and $\mathbf{g}_2$ are important, but DARTS chose to ignore $\mathbf{g}_2$ which can cause a dramatic error in approximation.

## A.2   THE COMPLEXITY OF $\mathbf{g}_2^{'}$ CAN BE SUBSTANTIALLY REDUCED USING THE FINITE DIFFERENCE APPROXIMATION

We use the finite difference approximation just like DARTS. Let $\epsilon$ be a small scalar, $\omega_1 = \omega + \epsilon \nabla_\omega \mathcal{L}_{\text{val}}(\omega, \alpha)$, $\omega_2 = \omega - \epsilon \nabla_\omega \mathcal{L}_{\text{val}}(\omega, \alpha)$. Then: $\mathbf{H} \cdot \nabla_\omega \mathcal{L}_{\text{val}}(\omega, \alpha) = \frac{\nabla_\omega \mathcal{L}_{\text{train}}(\omega_1, \alpha) - \nabla_\omega \mathcal{L}_{\text{train}}(\omega_2, \alpha)}{2\epsilon}$. $\omega_3 = \omega + \frac{\nabla_\omega \mathcal{L}_{\text{train}}(\omega_1, \alpha) - \nabla_\omega \mathcal{L}_{\text{train}}(\omega_2, \alpha)}{2}$, $\omega_4 = \omega - \frac{\nabla_\omega \mathcal{L}_{\text{train}}(\omega_1, \alpha) - \nabla_\omega \mathcal{L}_{\text{train}}(\omega_2, \alpha)}{2}$. Then: $\mathbf{g}_2^{'} = -\eta \times \frac{\nabla_\omega \mathcal{L}_{\text{val}}(\omega_3, \alpha) - \nabla_\omega \mathcal{L}_{\text{val}}(\omega_4, \alpha)}{2\epsilon}$

## A.3   A TOY EXAMPLE TO SHOW THE IMPORTANCE OF THE "GRADIENT TRAP"

We have a small toy case to show the influence of the "gradient trap". We searched for a small super-network in the DARTS's search space, which only has two cells(we searched for 600 epochs).

When we train the super-network in the "training sets in search phase" and validate in the "validation sets in search phase", the test error using $\mathbf{g}_2^{'}$ is $10.5\%$ while the test error without $\mathbf{g}_2^{'}$ is $12.8\%$.

When we train the super-network in the training sets and validate in the validation sets, the test error using $\mathbf{g}_2^{'}$ is $5.4\%$ while the test error without $\mathbf{g}_2^{'}$ is $7.4\%$.

When we generalize and train the network in the training sets and validate in the validation sets, the test error using $\mathbf{g}_2^{'}$ is $6.2\%$ while the test error without $\mathbf{g}_2^{'}$ is $7.4\%$.

In this case the "gradient trap" will cause a dramatic accuracy drop of the super-network.

## A.4   SEARCH WITH DIFFERENT SEEDS

We ran our search algorithms with different seeds for 5 times in $\mathcal{S}_2$ and evaluated each discovered architecture for 3 times. The lowest test error is $2.57\pm0.11\%$ and the highest is $2.63\pm0.13\%$. We did the same thing on $\mathcal{S}_1$ and the lowest and the highest test errors are $2.71\pm0.15\%$ and $2.92\pm0.09\%$, respectively.

As we expected, the results in $\mathcal{S}_1$ are less robust than those in $\mathcal{S}_2$. The main reason is the difference between search and evaluation, including the different depths of search and evaluation networks and the the discretization stage of the standard DARTS method.

More importantly, our approach survives after 500 (and even more) epochs, while DARTS degenerates to an all-skip-connect architecture in all (10+) individual runs.

## A.5   THEORETICAL ANALYSIS OF SEMI-POSITIVE-DEFINITE MATRIX AFTER SIMILARITY TRANSFORMATION

$\mathbf{A} = \mathbf{C}^T \cdot \mathbf{C}$, $\mathbf{A}$ is a semi-positive-definite matrix. In this case, the number of different eigenvalues is far less than the dimension of $\mathbf{A}$(hundreds compared to millions). $\mathbf{H}$ is a real symmetric positive-definite matrix.

Let $\{\alpha_i\}$ be a set of eigenvectors w.r.t $\mathbf{A}(\alpha_i^T \cdot \alpha_i = 1)$, then $\{\mathbf{H} \cdot \alpha_i\}$ is a set of eigenvectors w.r.t $\mathbf{H} \cdot \mathbf{A} \cdot \mathbf{H}^{-1}$, $\{\mathbf{H}^{-1} \cdot \alpha_i\}$ is a set of eigenvectors w.r.t $\mathbf{H}^{-1} \cdot \mathbf{A} \cdot \mathbf{H}$, and $\{\lambda_i\}$ is a set of eigenvectors shared by them.

Let $\beta$ be an eigenvector of $\mathbf{H} \cdot \mathbf{A} \cdot \mathbf{H}^{-1} + \mathbf{H}^{-1} \cdot \mathbf{A} \cdot \mathbf{H}$, $\beta = \sum_{i=1}^{n} a_i \times \mathbf{H} \cdot \alpha_i = \sum_{i=1}^{n} b_i \times \mathbf{H}^{-1} \cdot \alpha_i$.

$\left(\mathbf{H} \cdot \mathbf{A} \cdot \mathbf{H}^{-1} + \mathbf{H}^{-1} \cdot \mathbf{A} \cdot \mathbf{H}\right) \cdot \beta = \lambda\beta$, $\sum_{i=1}^{n} a_i\lambda_i \times \mathbf{H} \cdot \alpha_i + \sum_{i=1}^{n} b_i\lambda_i \times \mathbf{H}^{-1} \cdot \alpha_i = \sum_{i=1}^{n} a_i\lambda \times \mathbf{H} \cdot \alpha_i = \sum_{i=1}^{n} b_i\lambda \times \mathbf{H}^{-1} \cdot \alpha_i$.

$\sum_{i=1}^{n} (\lambda_i - \lambda) a_i b_i = -\sum_{i=1}^{n} (\lambda_i - \lambda) b_i \alpha_i^T \cdot \mathbf{H}^{-1} \cdot \sum_{i=1}^{n} (\lambda_i - \lambda) b_i \mathbf{H}^{-1} \cdot \alpha_i \leq 0$

A is a real symmetric matrix, so we have many sets of $\{\alpha_i\}$ that is orthogonal to each other. $\beta^T \cdot \beta = \sum_{i=1}^{n} a_i b_i \geq 0$, For every eigenvalue, the dimension of the subspace will be very high in the case of neural architecture search, so we assume that we can choose a set of $\{\alpha_i\}$ orthogonal to each other from the subspace satisfying $\sum_{j=n_i}^{n_{i+1}-1} a_j b_j \geq 0$ in most cases($n_i$ is the id of the first eigenvector w.r.t $\lambda_i$).

$\sum_{i=1}^{n} (\lambda_i - \lambda) a_i b_i = \sum_{i=1}^{n} (\lambda_i - \lambda) \sum_{j=n_i}^{n_{i+1}-1} a_j b_j \leq 0$, so $\lambda$ cannot be smaller than zero.

All of the eigenvalues w.r.t real symmetric matrix $\mathbf{H} \cdot \mathbf{A} \cdot \mathbf{H}^{-1} + \mathbf{H}^{-1} \cdot \mathbf{A} \cdot \mathbf{H}$ is not smaller than zero, so it is semi-positive-definite. Then we get that $\mathbf{H}^{-1} \cdot \mathbf{A} \cdot \mathbf{H}$ is semi-positive-definite.

