# OpenReview forum: "Stabilizing DARTS with Amended Gradient Estimation on Architectural Parameters"
_ICLR.cc/2020/Conference — Reject_

### Official Review · AnonReviewer4 · 2019-10-22
**Official Blind Review #4**

**Rating:** 3

**Review:**

The paper proposed to amend the 2nd order formulation of DARTS for improved stability. The main idea is to leverage the stationarity condition at a local optimum w* (that the derivative of the training loss equals zero). This is an interesting method, though the technique itself is not new outside the sub-field of architecture search. For instance, the same trick has been used in [1] for gradient-based hyperparameter optimization. Note the continuous architecture \alpha is mathematically equivalent to a high-dimensional hyperparameter.

Some major concerns:
* The authors correctly pointed out that the original 2nd-order approximation in DARTS is insufficient to make an accurate gradient direction. On the other hand, the proposed approximation in equation (8) seems aggressive enough to lead to outrageous approximation error. Specifically, the only guarantee is that the gradients before and after approximation form a non-negative angle (that their inner product is non-negative), whereas the angle alone can be insufficient to ensure the quality of individual elements, especially for high-dimension vectors.
* According to the authors, an advantage of the proposed approach is that one does not have to rely on early-stopping rules which require human expertise and “violates the fundamental ideology of AutoML” (Section 3.2). However, I am not fully convinced that the proposed method has an edge on this in practice, given the additional hyperparameter \eta (one may argue that those handcrafted early stopping rules are robust enough, just like the empirical statement in the paper that \eta = 0.1 worked well across the experiments) and potential approximation errors due to the Hessian.
* I'm also a bit concerned about the similar empirical performance but longer search time when comparing with other DARTS variants in Table 1 (using search space S1).

Minor issue:

In the introduction, the authors argue “convergence in search often leads to bad performance in re-training”, saying that a high validation accuracy during search is not a good indicator for the final performance. On the other hand, the goal of the proposed method is exactly to maximize the former rather than the latter. I believe this reasoning needs to be revised.

Question:
* Since each architecture gradient step is of comparable to the cost of 2nd order DARTS (which took 4 GPU days with 4 search repeats), it is not immediately clear why the proposed Amended-DARTS took only 1.1 GPU days (Table 1 & 2). Can you explain where did the speedup come from?
* Is there a particular reason to fix the edges in S1 and make it smaller than the original DARTS space? The question is relevant here because ideally we want an apple-to-apple comparison of those methods in the same space.

[1] Pedregosa, Fabian. "Hyperparameter optimization with approximate gradient.", ICML 2016

==== post-rebuttal comments ====
I would like to thank the authors for addressing some of my questions in the rebuttal. I decide to keep my score unchanged (weak reject). The reasons are as follows:
(1) Trying to improve DARTS from the optimization perspective is certainly interesting. However, the proposed approximation technique is rather a heuristic and only addresses the issue at a superficial level.
(2) Evaluating the proposed algorithm in the original DARTS space (in addition to the restricted space in the current manuscript) will substantially strengthen the paper.

**Experience Assessment:**

I have published one or two papers in this area.

**Review Assessment: Checking Correctness Of Derivations And Theory:**

I carefully checked the derivations and theory.

**Review Assessment: Checking Correctness Of Experiments:**

I carefully checked the experiments.

**Review Assessment: Thoroughness In Paper Reading:**

I read the paper thoroughly.

---

> ### Author Response · Authors · 2019-11-08
> **Responses to Reviewer #4**
>
> We thank the reviewer for the valuable comments. It seems that the reviewer did not realize the value of this work. The major contribution is to argue that *early-stop should be replaced by a more stable optimization method*, although it is a trick that produces reasonable results in the restricted search spaces. We delved deep into the optimization, found the error in computing the second-order term, and amended it with another term with a guaranteed property. Experiments verified that our method can produce strong architectures *under convergence* while existing methods all collapsed to the all-skip-connect failure.
>
> Before responding to the concerns, we would like to emphasize that the early-stop mechanism, as used by all DARTS-based approaches, is pushing NAS away from the correct direction. The reason is clear: with early-stop, (i) random initialization can largely impact search results and this is why the search algorithms are often unstable; (ii) it is difficult to generalize to a large search space (the main pursuit of NAS). We agree that our solution is temporary, but we tried to change the current status of research on DARTS by revealing a new direction. In one word, our approach pushes NAS research ahead, but early-stop seems not.
>
> Q0: The same trick has been used in [1].
>
> A0: Yes, the trick that derived H^-1 is not new, and our major contribution is to solve the problem that H^-1 is intractable due to high dimensionality.
>
> Q1: The proposed approximation has problems.
>
> A1: We agree that our approximation is 'aggressive' and the approximation error can be 'outrageous', however, to our best knowledge, this is the *only* existing approximation that the error is guaranteed. Indeed, it can be insufficient to ensure the quality of the approximation, but it is the best solution we have now. We empirically verified stable search results in (i) a very long search process and (ii) a very large search space, on which all existing DARTS-based methods failed dramatically.
>
> Q2: I am not fully convinced that the proposed method has an edge on this, given the additional hyper-parameter η and potential approximation errors.
>
> A2: Comparing a single hyper-parameter η to a carefully designed search mechanism is not fair, in particular, η is used for balancing loss terms and was not tuned carefully. Yes, the potential approximation errors are a problem, but this seems a more promising direction to work on, unlike the early-stop mechanism is only 'robust enough' in the restricted search spaces.
>
> Q3: The similar empirical performance but longer search time (Tab 1, space S1).
>
> A3: We need more time to train our model until convergence. Existing DARTS-based competitors all made an early-stop which saved search time, but we argue that their mechanism is wrong.
>
> Q4: A high validation accuracy during search is not a good indicator of the final performance?
>
> A4: In the original search space, a high validation accuracy is not *necessarily* good, because the pruning operation after search causes a significant gap between search and evaluation, and with this gap, optimizing towards the super-network *may* lead to an unsatisfying pruned sub-network (verified in experiments). We fixed the edges in the search space and maximally guaranteed the same settings of optimization (e.g. channel number, drop out rate, etc.) so that this gap was largely shrunk. In this case, it is very likely that a good super-network corresponds to a good sub-network (verified in experiments). So, fixing the edges is necessary for our approach. Please refer to Secs 3.5 and 4.1.2 for more details.
>
> Q5: Why did Amended-DARTS only 1.1 GPU-days?
>
> A5: There are a few differences between our approach and DARTS-2nd-order. We used more channels and layers, while DARTS-2nd-order had more edges and operations in each cell. Moreover, DARTS-2nd-order used the ꞷ value of the next step to replace ꞷ* while we used the current ꞷ*.
>
> Q6: Why fix the edges in S1? An apple-to-apple comparison in the same space?
>
> A6: Please refer to Q4 for the reason to fix the edges in S1. A fair comparison was made on the fixed-edge search spaces with the same hyper-parameters, and our approach outperforms DARTS significantly (Sec 4.1.3). Running our approach in the original DARTS space needs further technical efforts (e.g. a gradual pruning schedule), and we will study in the future.

---

### Official Review · AnonReviewer1 · 2019-10-23
**Official Blind Review #1**

**Rating:** 3

**Review:**

--- response after the author's rebuttal ---

Thank the authors to provide their response and let me clearly understand their contribution.

However, after considering those, I will not change my rating. The paper identifies the inaccurate gradient computation in the original DARTS and propose a new estimation and achieves constant improvement. While interesting, the experiments show this approach is just as good as some simple human tricks like early stopping. In their response, they also didn't intend to generalize their approach to other baselines, and it will significantly limit the impact of this work.

Authors insist that involving human prior is not the primary goal of AutoML. This is a subjective opinion, and my subjective opinion is it is wrong. If you take a closer look at all search spaces currently used in the NAS field, human priors are why NAS works up to now. For example, to use a convolutional operation, people use Conv-Bn-ReLU instead let NAS also search the activation functions. To search a cell rather than entire network is also designed by a human. If applied correctly, it is not wrong to include human priors.

As stated in guideline, we should apply a higher standard to a 10-page paper, and unfortunately, in my opinion, this paper does not meet such standard with the current version, and heavily revision will be necessary. I encourage the authors to shorten their paper by putting some of the derivations to the appendix and to better show their contribution. If not planning to generalize their method, show more ablation and toy example of gradient trap and how the approach solves that.

--- original review---


Summary

The paper presents a novel way to refactor second-order gradient term in Differentiable Architecture Search (DARTS) into the inverse of Hessian matrix (H^{-1}) corresponding to the optimal weight w^*, by leveraging a mathematical based on a property that gradient of loss w.r.t. w^* is 0. It provides an estimate relying on H instead of H^{-1} and showing mathematically that the proposed estimate has an angle (w.r.t ground-truth gradient vector) < 90 degrees, while the original estimate in DARTS is not bounded. The paper claims phenomenon (namely, gradient trap in DARTS) is the reason why DARTS and its successors constantly converge into a poor architecture while the proposed amendment will not.
It shows the experiments in both original DARTS search space with changes in the search phase, as well as a larger space with fewer operations.

The observation that DARTS converging to a fixed point is interesting and the motivation to find the reason and solution is well justified. The technical novelty to compute the actual and estimate term for second-order DARTS is okay, and the identified "gradient trap" seems reasonable. However, I have some doubts regarding both theoretical and empirical aspects of this paper. The evidences in the current version is not sufficient to support all the claims. Especially considering the 10-page length, I hope the author could clarify these concerns during the rebuttal period.

Major concerns

About the derivation

- Based on the derivation in Section 3.3, the proposed gradient amendment relies on Hessian Matrix regarding the optimum w^*(\alpha), while in reality, it is rarely satisfied. Will this brings another gradient trap? Could you comment on the theoretical bound between the proposed amendment and the original DARTS estimation? In addition, as stated in DARTS paper (Liu et al.) section 2.3, if w is already a local optimum, the \nabla L_train (w, \alpha) = 0, where this will also make g2 or g2' to 0.  However, to obtain a good estimate of the Hessian matrix, one will need to have a good estimate of w^*. Does this contradict the proposed approach?

- Computation of the Hessian matrix is missing, and the motivation to use g2'.
The proposed estimate depends on the Hessian matrix, however, it is not mentioned in the paper how to compute such matrix, especially "... is computationally intractable due to the large dimensionality of H, ... usually exceeds one million ..." as in Section 3.3. Also, authors proposed to use g2' on top of the Hessian matrix instead its inverse, could author add experiments to show the difference between g2 and g2'?

I came across to a concurrent paper submission [1], the derivation of second-order gradient estimate is the same as yours, i.e., one could use negative Hessian to improve the DARTS (see Appendix A.2 in [1]). However, the numerical computation of this Hessian is hard is also mentioned in [1]. Could the author also comment on that?

- Why the paper introduces \nabla_{\alpha} w^*(\alpha) with regarding L_train while the equation 2,3 is about validation loss? In DARTS, both training and validation dataset is used in training for better generalization.

About experiments

- Could the author provide an additional experiment, to show in reality, even for a toy example, this gradient trap make DARTS converge to an architecture full of skip-connection, i.e., showing the gradient estimate (i.e., g1+g2) w.r.t. \alpha, obtained by original DARTS, proposed approach, if possible, ground-truth estimate, after a certain epoch number (e.g. 50 or any number that DARTS start turning to skip-connection)? I think this will empirically reveals if the gradient trap is causing the DARTS problem and better demonstrate the paper's method effectiveness.

- Additional experiments for a fair comparison with baseline DARTS.
Experiment settings for both S1 and S2 seem to make the comparison to DARTS unfair. As shown in some previous work [2], different search space has different characteristic and is usually non-trivial in NAS domain. In addition, [3,4] shows that, weight sharing NAS is usually sensitive to random initialization, and results across runs can be quite different. To have a fair comparison, the paper should include additional experiments, running original DARTS and proposed one on the search space, S1 and S2, **with the same hyper-parameter setting**, for 3 runs with different random initialization. If the new results are still statistically significant, it will be strong evidence that the proposed algorithm indeed improves original DARTS.

Minor comments.

- Would the author kindly clarify the following points I found confusing during the reading?

1. Introduction third last paragraph: "Our final solution involves using the amended second term of [eq] brings \alpha meanwhile keeping the first term unchanged". What's does "brings \alpha?" means?
2. Section 3.3 line 7, is "fundamentals in mathematics" referring math foundations?

3. Equation 2, 3 are repetitive for no good reason?

4. In table 1,2 Random search results for S2 are missing. Could the author also listed for better comparison?

5. The second line after equation (3), "estimating \nabla ... ^2" why is this a square while in equation (3) it's not?


- generalization comparison with DARTS+, XNAS, etc.
Since the proposed gradient amendment is a general approach, adding this on top of other approaches seems natural. Could the author extend their method to other modified DARTS algorithms. It should obtain even better results with extra human-expertise after fixing the gradient trap. Adding this will further strengthen the paper.

- The writing is somehow informal and could be polished, e.g. "This is to say" in paragraph 1 of page 2, misuse of bold style, and some typos:
1. Introduction page 2, "In all experiments", 'I' should not be in bold.
2. line after equation 5, "throughout the remainder of this paper" remainder -> remaining.


--- Reference ---
[1] Anonymous, UNDERSTANDING AND ROBUSTIFYING DIFFERENTIABLE ARCHITECTURE SEARCH, link: https://openreview.net/forum?id=H1gDNyrKDS.
[2] Radosavovic et al., On Network Design Spaces for Visual Recognition, ICCV'19.
[3] Li and Talwalker, Random Search and Reproducibility for Neural Architecture Search, UAI'19.
[4] Scuito et al., Evaluating the search phase of neural architecture search, arxiv'19.

**Experience Assessment:**

I have published one or two papers in this area.

**Review Assessment: Checking Correctness Of Derivations And Theory:**

I carefully checked the derivations and theory.

**Review Assessment: Checking Correctness Of Experiments:**

I carefully checked the experiments.

**Review Assessment: Thoroughness In Paper Reading:**

I read the paper thoroughly.

---

> ### Author Response · Authors · 2019-11-08
> **Responses to Reviewer #1 (Part 1)**
>
> We thank the reviewer for the detailed comments. The main contribution of our work is to reveal the gradient trap in the two-stage optimization of DARTS and advocate for solving the essential problem rather than using early-stop to evade from it. This is a new direction which we believe is important. Although the reviewer raised several concerns, we believe most of them came from misunderstanding and can be easily addressed via further explanations.
>
> Q1: Based on the derivation... Does this contradict the proposed approach?
>
> A1: Let us explain the logic again. The accurate gradient computation (Eqn2) requires that the gradient of w* to be known. However, directly estimating this quantity often incurs a large approximation error (even if the accurate value of w* is known). This is the main reason that we used Eqn4 (originally Eqn5) to avoid computing it.
>
> On the other hand, the approximation error of Eqn2 can be very large if we ignore the second term of it. It may be a misunderstanding that "if w is already a local optimum, then \nabla [L_train(w,\alpha)]=0, where this will also make g2 or g2' to 0". In fact, DARTS used w'-\nabla [L_train(w,\alpha)] to replace w* before this statement. Although w=w* derives \nabla [L_train(w,\alpha)]=0, the gradient w.r.t. \alpha may not be 0 at this point. To show this, let us investigate a toy example here. Let the loss function be L(w,\alpha;x)=(x·w-\alpha)^2. Then, the only difference between L_train=L(w,\alpha;x_train) and L_valid = L(w,\alpha;x_valid) lies in the input, x. Assume the input of training data is x_train=1 and the input validation data is x_valid=2. It is easy to derive that the local optimum of L_train is w*(\alpha)=\alpha. Substituting x_valid=2 into L_valid yields L_valid=(2w-\alpha)^2. When \alpha=\alpha_t, w arrives at w*(\alpha_t), so g1=2(\alpha_t-2\alpha_t)=-2\alpha_t and g2=4\alpha_t. When w arrives at w*, L_valid(w*,\alpha)=\alpha^2, g(\alpha_t)=2\alpha_t=g1+g2. In summary, both g1 and g2 are important, but DARTS chose to ignore g2 which can cause a dramatic error in approximation.
>
> To bypass this error, we use the property of w*, and the result is that we can use H^-1 to compute the second term of Eqn2. Till here, no approximation was made. But, two problems remained. First, w may not arrive at the local optimum, for which we (as well as DARTS) naturally used w', the current status of w, as the best estimation of w* (this is almost the only thing we can do). Second, H^-1 is computationally intractable, for which we used H to replace H^-1. Although H itself is also intractable, we can estimate its projection on a certain direction by computing H*\nabla [L_val(w,\alpha)], where w' is used to replace w* (we simply followed DARTS).
>
> Back to the questions. DARTS-2nd-order used I to replace H^-1, and we used H. The angle between g2' (using H) and g2 (using H^-1) is not greater than 90 degrees, but there is no such guarantee between g2'' (using I) and g2 or between g2'' and g2'. We believe this property alleviates the gradient trap (see the next paragraph). The advantage of our approximation was verified in experiments.
>
> Let there be a loss function L(w). When w changes a little to w+dw, L changes to L(w+dw)≈\nabla L·dw accordingly. In gradient descent optimization, dw has the same direction as our estimation of -\nabla L to maximize the decrease of L. Thus, if our estimation of \nabla L and the accurate \nabla L have a positive inner-product, L is guaranteed to decrease in this iteration. This is why our optimization works better than that of the original DARTS.
>
> Q2: Computation of the Hessian matrix is missing... Could the author also comment on that?
>
> A2: Yes, H cannot be computed, but we can compute its projection on a certain direction. In this case, we do not need to compute H, but only H*\nabla [L_val(w,\alpha)]. The projection of H indicates the change of gradients in this direction, so we can compute the projection of H just like DARTS did.
>
> Note that there is no direct meaning of the projection of H^-1, which means we can calculate the projection of H but not of H^-1. This is the major difference between g2 and g2'. Since g2 (in particular, H^-1*\nabla [L_val(w,\alpha)]) cannot be computed, we cannot empirically show the difference between g2 and g2'. In [1], the authors derived g2 but find it intractable, so they moved back to the second-order method of DARTS. On the contrary, we moved one step forward and found a better approximation, g2', which is also easy to compute.
>
> [1] Anonymous, Understanding and Robustifying Differentiable Architecture Search, an ICLR 2020 Submission.
>
> (TO BE CONTINUED IN THE NEXT POST)

---

> > ### Comment · AnonReviewer1 · 2019-11-14
> > **Symbols in A1 A2 are missing; A2 is not clear**
> >
> > Thanks for your detailed response, and sorry for the late reply.
> >
> > For your response A1, I cannot see some of the symbols, some of them are just a blank square. Please revise your main text if possible, since I checked the latest revision does not contain any significant difference to read regarding this question.
> >
> > For your response A2, again some of the symbols are missing, but I could sort of grasp the ideas you would like to express.
> > To summarize, the technical contribution is to find a better estimation of gradients in DARTS, which relies on the formulation of g2'. However, this does not seem clear after reading your methodology many times. In the latest revision, your g2' (equation 6) still contains Hessian matrix H, while you claim you do not need to actually compute H.  "compute the projection of H just like DARTS" may not be enough to justify the formulation and for other people to implement your work.

---

> > > ### Author Response · Authors · 2019-11-14
> > > **The complexity can be substantially reduced using the finite difference approximation**
> > >
> > > A1:We revise our main text and we hope you could see the symbols.
> > > A2:We use the finite difference approximation just like DARTS. Let e be a small scalar, w1=w+e*\nabla [L_val(w,\alpha)], w2=w-e*\nabla [L_val(w,\alpha)]. Then:
> > > H*\nabla [L_val(w,\alpha)]=(\nabla [L_train(w1,\alpha)]-\nabla [L_train(w2,\alpha)])/2e
> > > w3=w+e*(\nabla [L_train(w1,\alpha)]-\nabla [L_train(w2,\alpha)])/2e, w4=w-e*(\nabla [L_train(w1,\alpha)]-\nabla [L_train(w2,\alpha)])/2e. Then:
> > > g2'=-eta*(\nabla [L_val(w3,\alpha)]-\nabla [L_val(w4,\alpha)])/2e

---

> > > > ### Comment · AnonReviewer1 · 2019-11-15
> > > > **response from reviewer**
> > > >
> > > > thanks for providing this. I think it is better to have this in main text to be self-contained. As said in earlier, I will make a final pass on the revised version after this discussion period.

---

> ### Author Response · Authors · 2019-11-08
> **Responses to Reviewer #1 (Part 2)**
>
> (CONTINUING)
> A2:We use the finite difference approximation just like DARTS. Let e be a small scalar, w1=w+e*\nabla [L_val(w,\alpha)], w2=w-e*\nabla [L_val(w,\alpha)]. Then:
> H*\nabla [L_val(w,\alpha)]=(\nabla [L_train(w1,\alpha)]-\nabla [L_train(w2,\alpha)])/2e
> w3=w+e*(\nabla [L_train(w1,\alpha)]-\nabla [L_train(w2,\alpha)])/2e, w4=w-e*(\nabla [L_train(w1,\alpha)]-\nabla [L_train(w2,\alpha)])/2e. Then:
> g2'=-eta*(\nabla [L_val(w3,\alpha)]-\nabla [L_val(w4,\alpha)])/2e
>
> Q3: Why the paper introduces...
>
> A3: Our formulation is the same as all DARTS-based methods, where the training and validation sets are made independent to avoid over-fitting. Please refer to the appendix of the original DARTS paper for detailed derivations.
>
> Q4: Could the author provide an additional experiment...
>
> A4: Even for a very small super-network in the DARTS search space, the number of parameters in w is over 4,000, and computing H^-1 in each step is intractable. But we have a small toy case to show the influence of "the gradient trap". We searched for a small super-network in the DARTS's search space, which only has two cells(we searched for 600 epochs).
> When we train the super-network in the "training sets in search phase" and validate in the "validation sets in search phase", the test error using g2' is 10.5% while the test error without g2' is 12.8%.
> When we train the super-network in the training sets and validate in the validation sets, the test error using g2' is 5.4% while the test error without g2' is 7.4%.
> When we generalize and train the network in the training sets and validate in the validation sets, the test error using g2' is 6.2% while the test error without g2' is 7.4%.
> In this case, the "gradient trap" will cause a dramatic accuracy drop of the super-network.
> In all the investigated search spaces (e.g. S1, S2, and 10+ others), DARTS consistently collapsed to all-skip-connect architectures after 200 epochs, yet our approach always produced reasonable architectures (and good results).
>
> Q5: Additional experiments for a fair comparison...
>
> A5: Thanks for this suggestion! We have run DARTS with the same hyper-parameters as used in our method, and the results were provided in the original submission (in the last paragraph of Section 4.1.3). We ran our search algorithms with different seeds for 5 times in S2 and evaluated each discovered architecture for 3 times. The lowest test error is 2.57±0.11% and the highest is 2.63±0.13%. We did the same thing on S1 and the lowest and the highest test errors are 2.71±0.15% and 2.92±0.09%, respectively.
>
> As we expected, the results in S1 are less robust than those in S2. The main reason is the difference between search and evaluation, including the different depths of search and evaluation networks and the the discretization stage of the standard DARTS method.
>
> More importantly, our approach survives after 500 (and even more) epochs, while DARTS degenerates to an all-skip-connect architecture in all (10+) individual runs.
>
> Q6: Generalization comparison with DARTS+, XNAS, etc.
>
> A6: This may be a misunderstanding. Our work aims to save DARTS from using human expertise, which, we believe, is not the right direction of NAS research. We find it very difficult to combine our approach with the manually designed tricks. For example, the success of DARTS+ was based on a hard constraint on the number of skip-connect operators, besides which it is the same as the original DARTS except for a few tricks. When considering a more complex search space like S2, the number of skip-connects dramatically varies for different cells. So it's hard to stop searching according to the number of skip-connects. XNAS introduced human expertise to DARTS, which is contradict to the motivation of AutoML (NAS).
>
> Minor concerns:
>
> 1) What does 'brings α' mean?
>
> A: Sorry for the typo: 'brings α' have been removed in the updated version.
>
> 2) Sec 3.3, Line 7,'fundamentals in mathematics'?
>
> A: Sorry for misleading. It means that the approximation error of DARTS-2nd-order cannot be bounded (the inner-product can be smaller than 0). We have clarified it in the updated paper.
>
> 3) Eqn2 and Eqn3 are repetitive?
>
> A: Yes, we repeated the same thing twice. We have removed the redundant contents.
>
> 4) Tabs 1 and 2: random search results for S2 are missing.
>
> A: Thanks for this suggestion! The random search results for S2 (worse than the amended search results) were provided in the last paragraph of Section 4.1.3. We do not list them in Tabs 1 and 2 because they are the places to compare against the state-of-the-arts.
>
> 5) The second line after Eqn3: why is this a square?
>
> A: The superscript '2' is not a power script but indicates a footnote. We have modified this part to avoid misunderstanding.
>
> Writing: Thanks! We have revised accordingly. BTW, 'I' was not in bold -- if we misunderstood anything, please point it out.

---

> > ### Comment · AnonReviewer1 · 2019-11-14
> > **Reviewer response**
> >
> > A3 is fine. My previous point saying both training and validation are used in "training", the "training" refers to the search phase not training the super-net.
> >
> > A4. Again, without a small toy case example, it is hard to conclude the "gradient trap" exists. Converging to a weird skip-connections can be caused by many factors, especially setting all operations to skip-connection does not mean the model having zero-parameter and should perform poorly during the search phase. It could also be this generalization gap between the search and evaluation phase of weight sharing NAS.
> >
> > Without proper demonstration or visualization, but only using the fact during search your algorithm does not converge to an "all skip-connection" model, is not enough to justify the existence of gradient gap and fixing it.
> >
> > A5.
> >
> > Thanks for the additional experiments. This shows a solid improvement over the DARTS baseline. Please merge this into your main paper with the missing citations in my first review.
> >
> > A6.
> >
> > Human expertise is important in AutoML. This is a subjective point and it is not worthy to spend time debating over it. However, my original concerns are, if you could fix the convergence issue in DARTS baseline, you should be able to do the same for other DARTS based methods as well. Otherwise, the generalization of your approach is not convincing by only showing an improvement over a simple baseline, while other similar tricks (even by human expertise) can get similar performances.
> >
> > Thanks for correcting the minor points.
> >
> > I also didn't see the bold "I".

---

> > > ### Author Response · Authors · 2019-11-14
> > > **A small toy case**
> > >
> > > A4:We still cannot calculate g2, but we have a small toy case to show the influence of "gradient trap". We searched for a small super-network in the DARTS's search space, which only has two cells(we searched for 600 epochs).
> > > When we train the super-network in the "training sets in search phase" and validate in the "validation sets in search phase", the test error using g2' is 10.5% while the test error without g2' is 12.8%.
> > > When we train the super-network in the training sets and validate in the validation sets, the test error using g2' is 5.4% while the test error without g2' is 7.4%.
> > > When we generalize and train the network in the training sets and validate in the validation sets, the test error using g2' is 6.2% while the test error without g2' is 7.4%.
> > > In this case the "gradient trap" will cause a dramatic accuracy drop of the super-network.

---

> > > > ### Comment · AnonReviewer1 · 2019-11-15
> > > > **interesting experiment**
> > > >
> > > > please update this in your main text together with other comments.
> > > >
> > > > I will make a final pass after the discussion period finished and provide my final evaluation.

---

> > > > > ### Author Response · Authors · 2019-11-15
> > > > > **We have updated our paper to be self-contained in the appendix**
> > > > >
> > > > > We have updated our paper and merged important comments using the appendix to our paper. We will try our best to merge them into the 10-page-length maintext in the final version!

---

### Official Review · AnonReviewer3 · 2019-10-29
**Official Blind Review #3**

**Rating:** 6

**Review:**

This paper offers through analysis on instability of NAS training that the system degenerates and reaches to trivial solutions, e.g., more skip-connect operators, as training goes longer. Motivated by this issue, this paper proposes an approach to stabilize NAS training.

Strength:
[1] Theoretical analysis
[2] The paper is well written
[3] This paper is trying to solve a very interesting and important problem

Weakness:
[1] Lack of ablation study. It would be better to show learning curve like Fig.1 to illustrate how the proposed approach helps to stabilize training and how about >500 epoch, any unstable?
[2]The proposed approach offers comparable results with SOTA (early stopping). I agree that it opens a direction of stable NAS training, but the contribution so far is limited. I expect to see quality gains due to improved training technology


**Experience Assessment:**

I do not know much about this area.

**Review Assessment: Checking Correctness Of Derivations And Theory:**

I did not assess the derivations or theory.

**Review Assessment: Checking Correctness Of Experiments:**

I carefully checked the experiments.

**Review Assessment: Thoroughness In Paper Reading:**

I read the paper at least twice and used my best judgement in assessing the paper.

---

> ### Author Response · Authors · 2019-11-08
> **Responses to Reviewer #3**
>
> We thank the reviewer for the positive comments.
>
> We agree that it is interesting and useful to provide an ablation study, showing that our approach indeed escapes from the gradient trap. We will add a new plot of our method in the DARTS search space to the updated version. Also, we describe here how the search process behaves under the stabilized DARTS algorithm. (i) The none ratio does not grow gradually to approach 1 but remains at a relatively low level close to 0.2. (ii) After 100 epochs, the weights of all operators change little. The entire architecture produces reasonably high accuracy and rarely changes.
>
> Regarding our contribution, we agree that it is sufficiently large as 'it opens a direction of stable NAS training'. We appreciate the suggestion of using 'improved training technologies' to improve the performance, and we will try the methods mentioned by [A], and report the results in the final version.
>
> [1] Anonymous, NAS Evaluation is Frustratingly Hard, an ICLR 2020 Submission.

---

### Official Review · AnonReviewer2 · 2019-10-31
**Official Blind Review #2**

**Rating:** 6

**Review:**

*UPDATE* I have read the other reviews, author's comments and the revised pdf. I maintain my weak accept rating, the paper is borderline but above the bar. The inclusion of experiments that Reviewer1 suggested definitely make the contributions stronger. I believe the paper will be substantially stronger with a careful study of where the empirical improvement is coming from. The theory (that the approximate gradient has an acute angle with the desired gradient) is potentially vulnerable -- this property is certainly desirable when stochastic-optimizing convex functions (with appropriate step sizes) but it's not trivial that it gives good behavior for optimizing non-convex functions like NAS. Without this careful study, it is not obvious that the proposed method doesn't suffer from it's own "gradient traps". That said, pointing out gradient approximation issues with differentiable NAS may be a valuable enough contribution.

The paper studies differentiable approaches to neural architecture search and convincingly points out that existing approximations to the gradient w.r.t. architecture parameters are problematic. A new approximation is proposed, and evaluated to show that degenerate architectures are not getting selected once search has converged (empirically) on standard image classification datasets.
The problem with existing approximations (e.g. first-order or second-order DARTS) is explained clearly. It is however unclear whether the proposed solution provides a complete solution, or if there are avenues for further improvement. The key question marks are: is the proposal a tractable approximation? is the empirical improvement indeed arising because of better gradients? Studying these two questions carefully via experiment will make the paper's contributions stronger.

Minor questions (related to writing/exposition):
How is Eqn2 different from Eq3? If they are the same, please remove the redundant equation (you already repeat it in the Introduction).
Why is Eqn7 a tractable approximation when H is very high-dimensional? I agree that it is more efficient that H^-1, but don't see how it can be tractable to compute in general.
Section 4.1.2: What is "auxiliary loss tower"?

Abstract: "obstacles it" (obstacles is an awkward verb, perhaps use "hinders it" instead)
Introduction: "was very few covered in previous works" is an awkward phrase. "has not been studied carefully in previous works"
Related work: "exhausted search" -> "exhaustive search"


**Experience Assessment:**

I do not know much about this area.

**Review Assessment: Checking Correctness Of Derivations And Theory:**

I assessed the sensibility of the derivations and theory.

**Review Assessment: Checking Correctness Of Experiments:**

I assessed the sensibility of the experiments.

**Review Assessment: Thoroughness In Paper Reading:**

I read the paper at least twice and used my best judgement in assessing the paper.

---

> ### Author Response · Authors · 2019-11-08
> **Responses to Reviewer #2**
>
> We thank the reviewer for the positive comments.
>
> Q1: It is unclear whether the proposed solution provides a complete solution, or if there are avenues for further improvement. The key question marks are: is the proposal a tractable approximation? Is the empirical improvement indeed arising because of better gradients?
>
> A1: We believe our paper provides a temporary solution, and we would not say it is complete and there is much room for future improvement, e.g. the approximation is not accurate enough. Nevertheless, to the best of our knowledge, this is the best approximation of DARTS till the date, as it guarantees that the estimated gradient and the true gradient (intractable) have an inner angle not greater than 90 degrees, while all other methods cannot. In addition, we empirically verify that our approach can achieve stable architectures when the search comes to convergence, but existing approaches mostly failed (collapsing to dumb architectures with all skip-connect operators).
>
> Q2: How is Eqn2 different from Eqn3? If they are the same, please remove the redundant equation.
>
> A2: Yes, they are the same. We will remove the redundant contents in the final version. Thanks!
>
> Q3: Why is Eqn7 a tractable approximation when H is very high-dimensional? I agree that it is more efficient than H^-1, but don't see how it can be tractable to compute in general.
>
> A3: Sorry for missing these details, because we simply followed what DARTS did. Indeed, Even H itself is computationally intractable, H*∇_ꞷ[L_val(ꞷ,α)] (its projection at a certain direction) indicates the change of gradients along this direction and can be easily estimated.
>
> Q4: Section 4.1.2: What is 'auxiliary loss tower'?
>
> A4: This is a standard optimization method proposed in [A]. The idea is to add a few auxiliary loss branches to the middle parts of a very deep network so that the gradient vanishing problem is alleviated in network training. It is also a standard optimization option used by all DARTS-based approaches.
>
> [A] Szegedy et al., Going Deeper with Convolutions, CVPR 2015.
>
> Other grammatical mistakes: Thanks, and we have fixed accordingly.

---

### Decision · Program_Chairs · 2019-12-19

**Decision:**

Reject

**Comment:**

This paper studies Differentiable Neural Architecture Search, focusing on a problem identified with the approximated gradient with respect to architectural parameters, and proposing an improved gradient estimation procedure. The authors claim that this alleviates the tendency of DARTS to collapse on degenerate architectures consisting of e.g. all skip connections, presently dealt with via early stopping.

Reviewers generally liked the theoretical contribution, but found the evidence insufficient to support the claims. Requests for experiments by R1 with matched hyperparameters were granted (and several reviewers felt this strengthened the submission), though relegated to an appendix, but after a lengthy discussion reviewers still felt the evidence was insufficient.

R1 also contended that the authors were overly dogmatic regarding "AutoML" -- that the early stopping heuristic was undesirable because of the additional human knowledge involved. I appreciate the sentiment but find this argument unconvincing -- while it is true that a great deal of human knowledge is still necessary to make architecture search work, the aim is certainly to develop fool-proof automatic methods.

As reviewers were still unsatisfied with the empirical investigation after revisions and found that the weight of the contribution was insufficient for a 10 page paper, I recommend rejection at this time, while encouraging the authors to take seriously the reviewers' requests for a systematic study of the source of the empirical gains in order to strengthen their paper for future submission.